# Investigating Predictive Factors of Dysphagia and Treatment Prolongation in Patients with Oral Cavity or Oropharyngeal Cancer Receiving Radiation Therapy Concurrently with Chemotherapy

Petros Alexidis [1], Pavlos Kolias [2], Vaia Mentesidou [3,*], Maria Topalidou [1], Efstathios Kamperis [1], Vasileios Giannouzakos [1], Konstantinos Efthymiadis [3], Petros Bangeas [4] and Eleni Timotheadou [5]

1 Radiation Oncologist, Department of Radiation Oncology, Papageorgiou Hospital, 56429 Thessaloniki, Greece
2 Section of Statistics and Operational Research, Department of Mathematics, Aristotle University of Thessaloniki, 56429 Thessaloniki, Greece
3 Medical Oncology Department, Aristotle University of Thessaloniki, Papageorgiou Hospital, 56429 Thessaloniki, Greece
4 1st University Surgery Department, Nanomedicine and Nanotechnology Aristotle University of Thessaloniki, Papageorgiou Hospital, 56429 Thessaloniki, Greece
5 Medical Oncologist, Medical Oncology Clinic Aristotle University of Thessaloniki, Papageorgiou Hospital, 56429 Thessaloniki, Greece
* Correspondence: vayamentesidou@hotmail.com

**Abstract:** Radiation therapy (RT) treatment for head and neck cancer has been associated with dysphagia manifestation leading to worse outcomes and decrease in life quality. In this study, we investigated factors leading to dysphagia and treatment prolongation in patients with primaries arising from oral cavity or oropharynx that were submitted to radiation therapy concurrently with chemotherapy. The records of patients with oral cavity or oropharyngeal cancer that received RT treatment to the primary and bilateral neck lymph nodes concurrently with chemotherapy were retrospectively reviewed. Logistic regression models were used to analyze the potential correlation between explanatory variables and the primary (dysphagia ≥ 2) and secondary (prolongation of total treatment duration ≥ 7 days) outcomes of interest. The Toxicity Criteria of the Radiation Therapy Oncology Group (RTOG) and the European Organization for Research and Treatment of Cancer (EORTC) were used to evaluate dysphagia. A total of 160 patients were included in the study. Age mean was 63.31 (SD = 8.24). Dysphagia grade ≥ 2 was observed in 76 (47.5%) patients, while 32 (20%) experienced treatment prolongation ≥ 7 days. The logistic regression analysis showed that the volume in the primary site of disease that received dose ≥ 60 Gy (≥118.75 cc, $p < 0.001$, (OR = 8.43, 95% CI [3.51–20.26]) and mean dose to the pharyngeal constrictor muscles > 40.6 Gy ($p < 0.001$, OR = 11.58, 95% CI [4.84–27.71]) were significantly associated with dysphagia grade ≥ 2. Treatment prolongation ≥ 7 days was predicted by higher age ($p = 0.007$, OR = 1.079, 95% CI [1.021–1.140]) and development of grade ≥ 2 dysphagia ($p = 0.005$, OR = 4.02, 95% CI [1.53–10.53]). In patients with oral cavity or oropharyngeal cancer that receive bilateral neck irradiation concurrently with chemotherapy, constrictors mean dose and the volume in the primary site receiving ≥ 60 Gy should be kept below 40.6 Gy and 118.75 cc, respectively, whenever possible. Elderly patients or those that are considered at high risk for dysphagia manifestation are more likely to experience treatment prolongation ≥ 7 days and they should be closely monitored during treatment course for nutritional support and pain management.

**Keywords:** dysphagia; radiotherapy; head; neck; cancer

## 1. Introduction

Head and neck cancer (HNC) is a commonly occurring malignancy accounting for 4–5% of cancer deaths, with 66,000 new cases diagnosed annually in the United States [1].



The 5-year overall survival for all head and neck cancers is estimated at 60% and is also influenced by other pathologies or comorbidities related to alcohol or tobacco abuse, such as lung or bladder cancer, second head and neck primaries, chronic obstructive pulmonary disease, and vascular disease. During recent years, there have been great advances in research including special patient populations such as young adults [2] or in the field of microbiome investigating the role of biomarkers in disease diagnosis, prognosis, or response to targeted therapies [3]. HNC is frequently treated with radiation therapy (RT) either in the definitive or postoperative setting. Despite the important evolution of radiation therapy techniques, RT related toxicity remains an important issue with great impact on the patients' quality of life and disease prognosis. Dysphagia is a commonly observed RT side-effect with an estimated prevalence of approximately 40% [4–6]. It is among the most important radiation therapy sequalae, since it has been associated with treatment breaks, therapy prolongation, health deterioration, and excessive weight loss [7–14]. Those factors frequently result in worse prognosis and quality of life. Multiple studies have highlighted the negative impact of treatment prolongation on prognosis, with an estimated 1–2% loss in local disease control for every single day of prolongation [10,15]. Moreover, weight loss [9,16,17] has been found to negatively affect treatment outcome. Skeletal muscle index has been identified as a predictive biomarker of worse overall and recurrence free survival [18], while weight loss above 10% after radiation therapy was associated with worse overall and disease specific survival [9]. Multiple studies have investigated the potential association of various factors with severe adverse reactions and malnutrition. Those factors include body mass index (BMI), age, primary site of disease, smoking or alcohol abuse, concurrent systemic therapy, and RT related parameters such as total dose, dose fractionation, and the extent of treated area [19–23]. It is frequently observed that patients presenting with primaries in the oral cavity or oropharynx (upper aerodigestive tract malignancies, UATM) are more likely to develop severe dysphagia [24–26], while bilateral neck irradiation and concurrent chemotherapy are considered strong predictors of higher RT related toxicity [20–23]. In this observational retrospective study, we are going to investigate factors predictive of dysphagia manifestation in patients with UATM primaries submitted to radiation therapy to the primary and bilateral neck concurrently with chemotherapy.

## 2. Materials and Methods

This study included head and neck cancer patients with primaries arising from the oral cavity or oropharynx that received volumetric arc radiation therapy (VMAT) directed to the primary site of disease and bilateral neck lymph nodes concurrently with chemotherapy. Both definitive and postoperative (adjuvant) treatments were allowed, while individuals were 18–85 years old, with stage I-IVB disease and World Health Organization (WHO) performance status 0–1. Only conventionally fractionated schemes were allowed, while patients were excluded if they met any of the following criteria: metastatic disease, palliative treatment, reirradiation, in-field recurrence after RT, prior RT in the head and neck region due to another primary carcinoma, postoperative treatment due to recurrence, no neck lymph node treatment, RT therapy other than VMAT (3D conformal or 2D), and alternate fractionation (hyper or hypofractionation).

The outcomes of interest included dysphagia manifestation (primary outcome) during or within 3 months from treatment completion and treatment prolongation (secondary outcome). Dysphagia was evaluated using the Toxicity Criteria of the Radiation Therapy Oncology Group (RTOG) and the European Organization for Research and Treatment of Cancer (EORTC) [27] and treatment prolongation was defined as any delay in treatment completion above 7 days since this has been associated with worse prognosis in previous reports [12,28].

Data regarding RT related parameters were extracted from the department's treatment planning software and included dose to the primary and lymph nodes, mean dose to the constrictor muscles, and the volume in the primary (PRvol) and neck (Nvol) that

received dose $\geq$ 60 Gy. Disease and population baseline characteristics were extracted from the department's electronic medical records and included age, dysphagia manifestation, treatment duration, treatment type (radical or postoperative), sex, and T and N stage.

The sample size was calculated with the intent to detect the joint effect of six predictor variables (outlined in the statistical analysis section) on the primary outcome of interest. We assumed that the prevalence of dysphagia grade $\geq$ 2 in the target population would be approximately 40%, thus by using the 10 events per variable (EPV) suggestion [29,30], the minimum sample size estimate was equal to 150 patients. The study was conducted in the General Hospital Papageorgiou of Thessaloniki, Greece, and approved by the institution's review board, while the need for patient informed consent was waived due to the study's retrospective design.

### 2.1. Treatment Preparation and Delivery

A 2 mm CT scan was initially acquired with the patient in supine position from the skull vertex to the sternoclavicular joints by using a thermoplastic mask to ensure patient immobilization. A bite block was used according to the physician's discretion to minimize the proximity of target volume to normal structures and reduce toxicity. Treatment was delivered daily, 5 days per week with 2 Gy/fraction, and the dose prescription was 66–70 Gy to the gross tumor volume, 6000–6105 Gy to the areas at high risk of harboring disease, and 5310–5412 Gy in the areas at low risk. Daily kilovoltage imaging was performed to verify patient positioning combined with a cone beam CT once weekly.

### 2.2. Statistical Analysis

Descriptive statistics for patient demographics and treatment or disease characteristics included means with standard deviation or medians with interquartile range according to the normality assumption for continuous variables, and frequencies with percentages for the categorical. Normality assumption was tested using the Shapiro–Wilk test. Both primary and secondary outcomes were analyzed as binary variables.

Dysphagia evaluation results were grouped into two categories (grade <2 vs. $\geq$2) as was treatment prolongation (<7 days vs. $\geq$7 days) and logistic regression models were used to investigate a potential correlation of predictor variable with the outcomes of interest. The variables included in the model were age, gender, therapy type, PR vol, N vol, and mean dose to the constrictors for the primary outcome and the same variables plus dysphagia manifestation for the secondary.

The area under the receiver operating characteristic curve (aROC) was used to assess the sensitivity and specificity of the model and the optimal cut-off probability was estimated with Youden index. The model's ability to discriminate between patients with or without severe toxicity was assessed using the C-statistic and Hosmer–Lemeshow goodness-of-fit test. Furthermore, Youden index was used to identify cut-off values for the constrictors means dosage and PRvol that could predict the primary outcome. The odds ratios (OR) for each predictor variable in the multivariable model, along with their 95% confidence intervals (95% CI) and *p*-values, were presented. The level of statistical significance was set to 0.05 for all tests. R studio v 1.4 was used for the analysis.

## 3. Results

Table 1 summarizes the cohort baseline and treatment characteristics. A total of 160 head and neck (HNC) patients treated with RT and concurrent chemotherapy between January 2018 and April 2021 were included. A total of 5 (3.12%) patients were stage I, 13 (8.12%) stage II, 37 stage III (23.12%), and 105 (65.62%) stage IV. A total of 92 individuals received postoperative RT (57.5%) vs. 68 (42.5%) that received radical RT. The mean treatment duration was 6.7 weeks, while 32 patients (20%) completed their treatment course with a prolongation $\geq$ 7 days. A total of 76 patients (47.5%) developed dysphagia grade $\geq$ 2. A total of 128 patients (80%) had positive lymph nodes (LN) and received $\geq$ 60 Gy to

the neck vs. 32 (20%) that were node negative. Median primary and neck volumes receiving ≥ 60 Gy were 120.46 cc (IQR = 53.88) and 153.61 cc (IQR = 138.43), respectively.

**Table 1.** Baseline characteristics.

| Characteristics | *n* | % |
|---|---|---|
| Gender | | |
| Female | 60 | 37.5 |
| Male | 100 | 62.5 |
| Age | | |
| Mean | | 63.31 |
| Standarddeviation | | 8.24 |
| Therapytype | | |
| Postoperative | 92 | 57.5 |
| Radical | 68 | 42.5 |
| Tstage | | |
| T1 | 5 | 3.12 |
| T2 | 13 | 8.12 |
| T3 | 61 | 38 |
| T4 | 81 | 50.76 |
| Nstage | | |
| N0 | 19 | 12 |
| N1 | 29 | 18 |
| N2 | 61 | 38 |
| N3 | 51 | 32 |
| Primarysubsite | | |
| Tongue | 55 | 60 |
| Floorofthemouth | 17 | 18 |
| Buccal | 6 | 6 |
| Retromolarspace | 9 | 10 |
| Lip | 6 | 6 |
| Baseoforopharynx | 29 | 43 |
| Tonsil | 22 | 32 |
| Softpalate | 13 | 20 |
| Pharyngealwall | 3 | 5 |
| Dysphagia | | |
| <2 | 84 | 52.5 |
| ≥2 | 76 | 47.5 |
| Treatmentprolongation | | |
| Yes | 32 | 20 |
| No | 128 | 80 |
| Constrictorsmean | | |
| Median | | 39.12 |
| IQR | | 10.49 |
| PRvol | | |
| Median | | 120.46 |
| IQR | | 53.88 |
| Nvol | | |
| Median | | 153.61 |
| IQR | | 138.43 |

PRvol: volume in the primary site receiving dose ≥ 60 Gy, Nvol: volume in the neck receiving dose ≥ 60 Gy, IQR: interquartile range.

Table 2 summarizes the univariate and multivariable regression models for the primary outcome of interest. Overall, the multivariable model showed adequate fit (C-statistic = 0.835, HL statistic = 4.81, $p = 0.5691$) and Youden index indicated a cut-off for specificity equal to

76%, which resulted in 79% sensitivity. In univariate analysis, constrictors mean ($p < 0.001$) and PR vol ($p < 0.001$) were statistically significant predictors of dysphagia $\geq 2$ manifestation. Both variables retained significance in multivariable analysis ($p < 0.001$). Specifically, patients with higher constrictors mean dose and higher volume in the primary area of disease treated to $\geq$60 Gy were more likely to develop dysphagia. We then sought to identify a clinically meaningful cut-off for each of these factors that could predict dysphagia manifestation by dichotomizing patients according to that value. Constrictor muscles (CM) mean dose and PRvol values were dichotomized by using the Youden index, which provided the thresholds that attained the best accuracy of the model. The two new binary variables were tested in a multivariable model to investigate the association with dysphagia by controlling the other covariates. Both CM mean $\geq 40.6$ Gy (OR = 11.58, 95% CI [4.84–27.71], $p < 0.001$) and PRvol $\geq 118.75$ cc (OR = 8.43, 95% CI [3.51–20.26], $p < 0.001$) were significant predictors of dysphagia manifestation.

**Table 2.** Univariate and multivariable logistic regression results for dysphagia grade $\geq 2$ development.

| Predictors | Univariate | | Multivariable | |
|---|---|---|---|---|
| | OR (95% C.I.) | *p* | OR (95% C.I.) | *p* |
| Gender | | | | |
| Female | | | | |
| Male | 1.05 (0.56–2) | 0.87 | 0.76 (0.367–1.578) | 0.463 |
| Age | 1.0032 (0.966–1.042) | 0.866 | 1.01 (0.965–1.054) | 0.699 |
| Therapy type | | | | |
| Postoperative | | | | |
| Radical | 1.46 (0.78–2.75) | 0.237 | 1.255 (0.621–2.536) | 0.528 |
| Constrictors mean | 1.07 (1.04–1.11) | **<0.001** | 1.081 (1.042–1.122) | **<0.001** |
| PRvol | 1.01 (1–1.02) | **<0.001** | 1.012 (1.005–1.019) | **<0.001** |
| Neckvol | 1.000 (0.998–1.003) | 0.746 | 1.000 (0.998–1.003) | 0.769 |

PRvol: volume in the primary site receiving dose $\geq 60$ Gy, Nvol: volume in the neck receiving dose $\geq 60$ Gy. The bold designates which comparisons were statistically significant.

Regarding the secondary outcome (Table 3), both univariate and multivariable models identified age and dysphagia grade $\geq 2$ as significant predictors of treatment prolongation $\geq 7$ days ($p = 0.007$ and 0.005, respectively).

**Table 3.** Univariate and multivariable logistic regression results for treatment prolongation $\geq 7$ days.

| Predictors | Univariate | | Multivariable | |
|---|---|---|---|---|
| | OR (95% C.I.) | *p* | OR (95% C.I.) | *p* |
| Gender | | | | |
| Female | | | | |
| Male | 1.18 (0.53–2.67) | 0.683 | 1.31 (0.55–3.14) | 0.544 |
| Age | 1.06 (1.01–1.12) | **0.016** | 1.079 (1.021–1.140) | **0.007** |
| Therapy type | | | | |
| Postoperative | | | | |
| Radical | 1.71 (0.78–3.73) | 0.177 | 1.76 (0.76–4.11) | 0.188 |
| Constrictors mean | 1.01 (0.97–1.05) | 0.565 | 1.034 (0.992–1.078) | 0.112 |
| PRvol | 1.000 (0.993–1.006) | 0.952 | 1.004 (0.996–1.012) | 0.317 |
| Neckvol | 1.001 (0.998–1.004) | 0.533 | 1.002 (0.999–1.005) | 0.247 |
| Dysphagia $\geq 2$ | 2.53 (1.13–5.69) | **0.024** | 4.02 (1.53–10.53) | **0.005** |

PRvol: volume in the primary site receiving dose $\geq 60$ Gy, Nvol: volume in the neck receiving dose $\geq 60$ Gy. The bold designates which comparisons were statistically significant.

## 4. Discussion

In this study, we found that radiation therapy induced dysphagia in patients with UATM primaries treated with concurrent chemotherapy and bilateral neck irradiation is associated with mean dose to the constrictors and to the volume in the primary site of disease that is treated with dose $\geq 60$ Gy. Moreover, we found that older patients or those

that developed dysphagia grade $\geq 2$ were more likely to receive a prolonged treatment by $\geq 7$ days.

We did not find an association between age and the primary endpoint of interest. Age has been identified as a predictive factor of increased toxicity, frequently resulting in decreased treatment tolerance in various clinical scenarios including head and neck cancer patients [20,21]. Although increased age was not predictive of acute dysphagia, it was significantly correlated with treatment prolongation, implying that elderly patients are more vulnerable to aggressive therapies and should be closely monitored during treatment course for early intervention and supportive care. Type of therapy was not predictive of either dysphagia or treatment prolongation. That was an interesting finding since definitive and postoperative treatment represent different therapeutic approaches, with sources in the literature demonstrating mixed results regarding their toxicity profiles [13,14,31,32]. Surgical interventions in combination with subsequent RT or chemo-RT frequently result in higher side effects [31,32]. On the other hand, higher radiotherapy doses delivered during definitive treatment have been associated with worse toxicity manifestation compared to the postoperative setting [13,14]. It is unclear whether higher rates of side effects should be expected during the definitive or postoperative setting, and this has been verified by our study. Based on our results, more attention should be paid to dose distribution to normal tissues (expressed by mean dose to the constrictor muscles and the volume of the treatment are in the primary site of disease), rather to the type of treatment.

Patients treated in the oral cavity or oropharynx constitute a special subgroup of head and neck cancer population since they frequently present with higher rates of dysphagia, malnutrition, and excessive weight loss [24–26]. A study by Bosh et al. [24] aimed to develop normal tissue complication probability (NTCP) models for 22 types of toxicity manifestation (including swallowing, mucosal irritation, speech, and pain) by extracting data from 750 head and neck patients submitted to radiation therapy. The relationship between normal tissue irradiation and toxicities was investigated in 10 different time points. In multivariate analysis, oral cavity was the predominant predictor associated with 12 toxicities. Buccal mucosa, which is an anatomical subsite of oral cavity, was also a strong predictor of various toxicities including swallowing difficulties. Another study [26] retrospectively explored the predictive role of patient and treatment characteristics for weight loss in 476 head and neck cancer patients who received radiation therapy either in definitive or postoperative setting. Pre-treatment body mass index (BMI), disease stage, and primary site of disease statistically predicted weight loss, with primaries arising from the oral cavity or oropharynx being associated with 3.86 higher odds of more than 10% weight loss (95% CI: 1.73–8.61; $p = 0.001$).

The negative prognostic value of weight loss during therapy has been verified in numerous studies [8–11]. It is believed that this is the result of events linked to weight loss, such as weakening of the organism, immune system deficiency, health deterioration, and treatment alterations including prolongation of therapy completion, interruptions, and deintensification of initially planned treatment schedule [8,9,15–18]. Weight loss has also been associated with decreased capability to tolerate optimal treatment and increased rates of chemotherapy induced toxicity [10,11,15] and postoperative complications [8]. A retrospective analysis of 246 head and neck patients receiving concurrent chemoradiation investigated the role of sarcopenia in disease prognosis [17]. A total of 37% of patients experienced treatment delays of >1 week and 14% had treatment breaks of more than 1 week. Sarcopenia was statistically associated with longer treatment breaks, increased chemotherapy derived toxicity, and worse overall survival (OS) (HR 1.83, $p = 0.03$) and progression free survival (PFS) (HR 1.65, $p = 0.03$). Moreover, a study including 1340 patients diagnosed and treated for head and neck cancer [9] found that critical weight loss (WL) (defined as >5% weight loss during radiation therapy or >7.5% until week 12) was associated with worse OS and disease specific survival (DSS). More specifically, the 5-year OS and DSS were 62% and 82% for patients with critical WL vs. 70% and 89% for those with no critical WL ($p = 0.01$, $p = 0.001$). WL > 10% before RT was also a strong predictor of worse

OS after adjusting for potential confounding. Being able to predict which patients are at higher risk of dysphagia and weight loss would further facilitate individualized care and precautionary measures. In this study, we identified a cut-off for mean dose (40.6 Gy) to the constrictors and for the volume in the primary site that receives $\geq$ 60 Gy (118.75 cc) that were both associated with increased rates of grade $\geq$ 2 dysphagia. Based on our findings, values should be kept below those limits whenever possible.

Treatment prolongation investigated in this study has been identified as an important prognostic factor of worse survival in multiple reports. Xiang et al. [28] performed a large analysis of 36,367 patients from the National Cancer Database with the intent to investigate a potential impact of treatment prolongation of definitive radiation therapy on survival. Treatment prolongation was found to negatively affect survival by increasing the risk of death by 2% per day ($p < 0.001$). Moreover, patients with treatment prolongation of >8 days had 12% lower overall survival at 4 years compared to those with a prolongation of 1–3 days. Another retrospective study reported the results on the tolerance of definitive RT combined with cetuximab in patients with squamous carcinomas of the head and neck [33]. Severe mucositis was observed in 57.6% of patients and 53% had to temporarily discontinue the treatment due to acute side-effects. In multivariate analysis, treatment discontinuation was predictive of survival, with each lost day of treatment resulting in an increase by 2% in the relative hazard of death. Similarly, a National Cancer Data Base study including 19,531 patients investigated the effect of radiation treatment time (RTT) on OS in head and neck patients receiving RT with or without concurrent chemotherapy in both the definitive and postoperative setting. Prolonged RT (defined as >56 and >49 days for definitive and postoperative RT, respectively) was associated with worse survival and the same observation was also confirmed for the 9200 patients that received concurrent chemotherapy [34]. Regarding local control, two studies observed higher rates of local failure when treatment was extended > 7 days [11,12]. We found that treatment prolongation $\geq$ 7 days was more likely to occur in elderly patients and those experiencing grade $\geq$ 2 dysphagia. Adequate management of side-effects in those patients deemed at higher risk for dysphagia manifestation and subsequent treatment prolongation is of critical importance and they should be closely monitored for early intervention. Intervention could include administration of opioids at pain onset, gabapentin to treat neuropathic pain [35], oral mouthwashes, dietary instructions, and nutritional support such as nasogastric feeding tube or gastrostomy (GT).

Specifically for gastrostomy, there are conflicting data in the literature regarding the optimal timing of GT usage, with some sources favoring prophylactic GT [36,37] (placement before start of treatment when higher adverse reactions are expected) and other favoring reactive GT [38,39] (placement during treatment in those patients manifesting severe toxicity). There are no randomized trials comparing the two methods, and the scarce data come mainly from retrospective series. According to a systematic review [39], pPEG significantly decreased malnutrition and was correlated with superior quality of life 6 months after treatment completion, while no difference was observed in average weight loss at different time points between pPEG and rPEG. Tumor control and overall survival were not affected by the type of gastrostomy, and pPEG resulted in longer PEG dependence. Similarly, other data demonstrated that pPEG improved nutrition was associated with lower rates of aspiration, strictures, or hospital admissions but was combined with worse long term swallowing outcomes and longer gastrostomy dependence [37]. Contradictory results were observed in other reports, which found no benefit of pPEG in weight loss, therapy toleration, or treatment alterations and outcomes [38,39]. Although many institutes are shifting towards reactive usage [40], prophylactic gastrostomy remains a valid option for those patients considered at high risk for dysphagia development and our study's results could guide decisions in those cases.

In this study, we did not find a correlation between dysphagia or treatment prolongation with the volume in the lymph nodes of the neck that was treated to dose $\geq$ 60 Gy, although it is reasonable to expect higher rates of adverse events when greater RT dose is delivered to larger anatomical areas. A possible explanation for this could be that it

is not just the extent of the area, but also the topography of the anatomical subsites that need to be treated with higher doses. When higher RT dose is delivered to lymph nodes that are in close proximity to critical anatomical structures involved in swallowing (e.g., constrictors muscles or oral cavity), the impact on the toxicity profile might be greater compared to the anatomical volume that receives the corresponding dose. A study by Wentzel et al. [41], which investigated the potential correlation of toxicity manifestation and lymph node spread pattern of the disease, verified that the topography of the disease in the neck plays a role in the toxicity profile. More specifically, they found that patients with disease in bilateral lymph node stations 2A,B and 3 were more likely to develop adverse events compared with bilateral 2A,B and unilateral 3 or unilateral 2A,B, 3, and 4. The outcome of interest was also affected by the extent of disease in the LN stations. Further investigation on this topic would help identify which patients are at higher risk of adverse reactions based on the pattern of disease spread in their lymph node stations.

The study's limitations are, first, its retrospective design. Limitations frequently observed in such studies include confounding, unbalanced population characteristics, and various types of bias (recall, misclassification, or selection bias) that could have an impact on final outcomes. Another limitation is that we did not distinguish between chemotherapy type or agents. This could have introduced confounding in the study, since there is evidence indicating differences in the toxicity profile between various chemotherapeutic agents or induction chemotherapy followed by concurrent chemo-RT compared to concurrent chemo-RT alone. Finally, the sample size was calculated with the intent to detect the effect of a specific number of explanatory variables on the primary outcome of interest. The final sample size was not large enough to allow for the inclusion of other clinically significant variables that would be interesting to investigate such as WHO performance status, weight loss during therapy, type of chemotherapy, or different anatomical subsites of oral cavity and oropharynx. Similarly, we could not investigate the survival of our patients and factors related to it such as other comorbidities, history of tobacco and alcohol abuse, and other primaries such as lung or bladder cancer.

## 5. Conclusions

In this study, we identified CM mean dose > 40.6 Gy and PRvol $\geq$ 118.75 cc as significant predictors of dysphagia in patients with UATM malignancies receiving concurrent radiotherapy–chemotherapy. We also found that patients developing dysphagia are more likely to experience treatment prolongation which can have detrimental effect on disease prognosis. Our findings could facilitate individualized care in those patients considered at higher risk for adverse reactions.

**Author Contributions:** P.A. was responsible for designing and supervising the study, collecting the data, and writing the manuscript. P.K. was responsible for analyzing the data and interpreting the results. V.M., E.K., M.T., V.G., P.B., K.E. and E.T. contributed to writing the manuscript. All authors have read and agreed to the published version of the manuscript.

**Funding:** This research received no external funding.

**Institutional Review Board Statement:** The study was conducted in accordance with the Declaration of Helsinki, and approved by the Institutional Review Board of General Hospital Papageorgiou, Greece (758/23-10-2020). Ethical review and approval were waived for this study due to the study's retrospective design.

**Informed Consent Statement:** Due to the study's retrospective design, the need for ethics approval and consent to participate was waived. The authors did, however, receive approval from the hospital's ethics committee to gain access to the HNC patient files and electronic database: 758/23-10-2020. The study was approved by the review board of General Hospital Papageorgiou, Thessaloniki, Greece.

**Data Availability Statement:** The datasets used and/or analyzed during the current study are available from the corresponding author upon reasonable request.

**Conflicts of Interest:** The authors declare no conflict of interest.

## Abbreviations

| | |
|---|---|
| CI | confidence interval |
| CM | constrictor muscles |
| EPV | events per variable |
| GT | gastrostomy |
| HNC | head and neck cancer |
| IQR | interquartile range |
| LN | lymph nodes |
| Nvol | neck volume receiving dose $\geq 60$ Gy |
| OR | odds ratio |
| Prvol | primary site volume receiving dose $\geq 60$ Gy |
| ROC | receiver operating characteristics |
| RT | radiation therapy |
| RTOG | radiation therapy oncology group |
| UATM | upper aerodigestive tract malignancies |
| VMAT | volumetric arc therapy |
| WHO | World Health Organization |

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
