# Peer review of "Investigating Predictive Factors of Dysphagia and Treatment Prolongation in Patients with Oral Cavity or Oropharyngeal Cancer Receiving Radiation Therapy Concurrently with Chemotherapy"

_curroncol, doi:10.3390/curroncol30050391_

Round 1
Reviewer 1 Report
The article has a practical importance and the conclusions should point more clearly some recommendation based on the results.
It seems that there are no significant differences between postoperative radiotherapy and radical chemoradiation, as shown in the results table. It will be of help if authors add a comment on factors that are shown in the table as not having significant impact on developing dysphagia or delayed treatment.
Overall, I consider it worthwhile for publication, being of interest for practice.

Author Response
Please see the attachment
Attachment File:
Dear reviewer,
Thank you very much for your comments.
We have added a paragraph in discussion section commenting on those factors.
Reviewer 2 Report
Well conducted study for serious issue like dysphagia during and after radiotherapy as radical or post operative setting. Dysphagia is multifactorial in such patients and finding out one cause may not be sufficient enoughd for different groups of patients who are suffering from head and neck malignancies..
Author Response
please see the attachment
Attachment File:
Dear reviewer,
Thank you very much for your comments.
We totally agree with your statement that dysphagia is a multifactorial issue and identifying more predictive factors would further facilitate clinical practice. Unfortunately, the study’s sample size did not allow for the inclusion of more predictive variables that would be of clinical importance and this is mentioned in the limitations paragraph.
Inclusion of variables like induction chemotherapy, type of chemotherapy agents or different subsites of oral cavity and oropharynx could have yielded more interesting results, but the study was not powered for that.
Nevertheless, we identified some factors that we believe could offer guidance in the management of patients with head and neck malignancies.
It is our intention to conduct a larger study in the close future including a larger number of predictive variables to further investigate the important issue of dysphagia in HNC patients.
Reviewer 3 Report
The topic is important and timely. I have read with great interest the paper and I congratulate Authors. In my opinion, in the introduction, the importance of advances in research as well as treatment in the young adults should be stressed. Therefore, introduction would be enhanced by addition of references, such as PMID: 31683170 and PMID: 27933385 to better contextualize the issue at hand in oncologic scenario.
It is not clear the RT scheme. Authors declared “Only conventional fractionation RT schemes were allowed (2 Gy/fraction)” (see line 74) but actually they prescribed “6000 – 6105 Gy to the areas at high risk of harboring disease and 5310 – 5412 Gy in the areas at low risk.” (see lines 106-107). Please explain it.
Please add in Table tumor characteristics (T stage, N stage, primary subsite). Considering that in locally advanced disease the standard of care is concomitant CHT, one can argue if the reported results are still representative for the toxic value. It is not clear (at least to me) how such problems were statistically tackled with.
Author Response
please see the attachment
Attachment File:
Dear reviewer,
Thank you very much for your comments.
Reply to comment “In the introduction, the importance of advances in research as well as treatment in the young adults should be stressed”: We have added comments in the introduction concerning the role of microbiome in head and neck cancer and the treatment of young adults, with the addition of the suggested references.
Reply to the comment “It is not clear the RT scheme. Authors declared “Only conventional fractionation RT schemes were allowed (2 Gy/fraction)” (see line 74) but actually they prescribed “6000 – 6105 Gy to the areas at high risk of harboring disease and 5310 – 5412 Gy in the areas at low risk.” (See lines 106-107). Please explain it.”: We agree that the way it is written in the text could be confusing for the reader. This study excluded patients who received either hyper or hypofractionated RT regimens. The conventional 2 Gy daily fractionation refers to the dose delivered to the primary site of disease. Since the treatment was delivered with simultaneous integrated boost (SIB) the daily dose to areas of high and low risk of harboring disease ranged between 1,8 – 1,9 Gy and 1,6 – 1,75 Gy respectively. To correct this confusion, we are going to rephrase to “Only conventionally fractionated RT schemes were allowed.’’
Reply to comment “please add in table tumor characteristics (T stage, N stage, primary subsite)”: We have added the aforementioned information in table 1.
Reply to comment “Considering that in locally advanced disease the standard of care is concomitant CHT one can argue if the reported results are still representative for the toxic value”: Our intention was to investigate predictive factors of dysphagia and treatment prolongation in head and neck cancer patients who are already at higher risk of toxicity manifestation. This was reflected in our inclusion criteria regarding the primary site of disease (oral cavity and oropharynx), the extend of irradiation fields (bilateral neck irradiation) and the use of systemic therapy (concomitant use of chemotherapy was obligatory). There are multiple sources in the literature establishing that all those factors have been associated with higher rates of adverse events. In that sense, the results of this study are not fully representative for all the head and neck cancer (HNC) scenarios but are focused on that HNC population which is at higher risk for toxicity. The aim of our research was to identify factors that can guide decisions and precautionary measures for this specific subgroup of HNC patients.
Reviewer 4 Report
Dear Author, I have carefully read your interesting study evaluating the risk factors for dysphagia and prolonged hospitalization in patients with cancer of the oral cavity or oropharynx undergoing chemoradiotherapy.
Here are my comments:
- The dysphagia assessment scale could be specified in the abstract. Its citation is not clear even when explained in materials and methods (lines 83-84), you could cite [Cox JD, Stetz J, Pajak TF. Toxicity criteria of the Radiation Therapy Oncology Group (RTOG) and the European Organization for Research and Treatment of Cancer (EORTC). Int J Radiat Oncol Biol Phys. 1995 Mar 30;31(5):1341-6 . doi: 10.1016/0360-3016(95)00060-C. PMID: 7713792]
- The introduction could further develop the discussion of dysphagia. What is the frequency of post RT/chemo-RT dysphagia in the literature? What is the impact for the patient and for the healthcare system? Are there articles assessing risk factors for dysphagia or is there still little evidence? [Jiang N, Zhang LJ, Li LY, Zhao Y, Eisele DW. Risk factors for late dysphagia after (chemo)radiotherapy for head and neck cancer: A systematic methodological review. Head Neck. 2016 May;38(5):792-800. doi: 10.1002/hed.23963. Epub 2015 Jun 18. PMID: 25532723.]
- The materials and methods section could specify from which center the patients were selected.
- You’ve assumed that the prevalence of dysphagia is 40%, is there any literature in favor of that or does this frequency comes out from the experience in your center? How were the patients selected? Are they consecutive patients within the database?
- In the presentation, as well as in table 2, you talk about constrictor mean, do you mean constrictor mean dose?
Author Response
please see the attachment
Attachment File:
Dear reviewer,
Thank you very much for your comments.
Reply to comment “The dysphagia assessment scale could be specified in the abstract. Its citation is not clear even when explained in materials and methods (lines 83-84), you could cite [Cox JD, Stetz J, Pajak TF. Toxicity criteria of the Radiation Therapy Oncology Group (RTOG) and the European Organization for Research and Treatment of Cancer (EORTC). Int J Radiat Oncol Biol Phys. 1995 Mar 30;31(5):1341-6 . doi: 10.1016/0360-3016(95)00060-C. PMID: 7713792]”: We have added a comment about the assessment scale in the abstract, rephrased our reference the RTOG scale in the materials and methods and added a citation.
Reply to comment “The introduction could further develop the discussion of dysphagia. What is the frequency of post RT/chemo-RT dysphagia in the literature? What is the impact for the patient and for the healthcare system? Are there articles assessing risk factors for dysphagia or is there still little evidence? [Jiang N, Zhang LJ, Li LY, Zhao Y, Eisele DW. Risk factors for late dysphagia after (chemo)radiotherapy for head and neck cancer: A systematic methodological review. Head Neck. 2016 May;38(5):792-800. doi: 10.1002/hed.23963. Epub 2015 Jun 18. PMID: 25532723.]”: We further developed the issue of dysphagia in the introduction, presented the estimated rates of dysphagia in head and neck cancer patients submitted to radiotherapy and referred to articles in the literature that have investigated risk factors of dysphagia.
Reply to comment” the materials and methods section could specify from which center the patients were selected”: We have specified our center in the materials and methods.
Reply to comment “You’ve assumed that the prevalence of dysphagia is 40%, is there any literature in favor of that or does this frequency comes out from the experience in your center? How were the patients selected? Are they consecutive patients within the database?”: We assumed 40 % prevalence of dysphagia based on data in the literature (sources outlined below)which are consistent with our own experience in our center.
- Dysphagia in head and neck cancer, Kapila Manikantan et al. Cancer Treat Rev. 2009Dec;35(8):724-32.
- Two-year prevalence of dysphagia and related outcomes in head and neck cancer survivors: An updated SEER-Medicare analysis, Katherine A Hutcheson et al. Head Neck. 2019Feb;41(2):479-487.
- Long-term prevalence of oropharyngeal dysphagia in head and neck cancer patients: Impact on quality of life, P García-Peris et al. Clin Nutr. 2007 Dec;26(6):710-7
Reply to comment” how were the patients selected? Are they consecutive within the database?”: The patients were selected based on the study’s inclusion and exclusion criteria outlined in the materials and methods section. They were not consecutive within the database, but they all received treatment during the sameperiod of time (from 2018 until 2021).
Reply to comment “in table 2 you talk about the constrictor mean, do you mean constrictor mean dose?”: Yes we mean the mean dose to the constrictor muscles.
Round 2
Reviewer 3 Report
revision ok
Reviewer 4 Report
Dear authors, thank you for responding to my observations. However, the captions of tables 2 and 3 explain the meaning of "Nvol", while "NeckVol" is reported in the tables. Just as it should be specified that "constrictors mean" is a dose.